# Plant species determine tidal wetland methane response to sea level rise

Peter Mueller [1,2✉], Thomas J. Mozdzer[3], J. Adam Langley [4], Lillian R. Aoki [5], Genevieve L. Noyce [1] &
J. Patrick Megonigal [1✉]

Blue carbon (C) ecosystems are among the most effective C sinks of the biosphere, but
methane ($CH_4$) emissions can offset their climate cooling effect. Drivers of $CH_4$ emissions
from blue C ecosystems and effects of global change are poorly understood. Here we test for
the effects of sea level rise (SLR) and its interactions with elevated atmospheric $CO_2$,
eutrophication, and plant community composition on $CH_4$ emissions from an estuarine tidal
wetland. Changes in $CH_4$ emissions with SLR are primarily mediated by shifts in plant
community composition and associated plant traits that determine both the direction and
magnitude of SLR effects on $CH_4$ emissions. We furthermore show strong stimulation of $CH_4$
emissions by elevated atmospheric $CO_2$, whereas effects of eutrophication are not significant.
Overall, our findings demonstrate a high sensitivity of $CH_4$ emissions to global change with
important implications for modeling greenhouse-gas dynamics of blue C ecosystems.

[1] Smithsonian Environmental Research Center, Edgewater, MD 21037, USA. [2] Institute of Soil Science, Center for Earth System Research and Sustainability
(CEN), Universität Hamburg, 20146 Hamburg, Germany. [3] Department of Biology, Bryn Mawr College, Bryn Mawr, PA 19010, USA. [4] Department of Biology,
Center for Biodiversity and Ecosystem Stewardship, Villanova University, Villanova, PA 19003, USA. [5] Department of Ecology and Evolutionary Biology,
Cornell University, Ithaca, NY 14853, USA. ✉email: peter.mueller@uni-hamburg.de; megonigalp@si.edu

Tidal wetlands (i.e. marshes and mangroves) are often characterized by lower emissions of the powerful greenhouse gas $CH_4$ than nontidal wetlands[1–4]. Microbial $CH_4$ production in wetland soils is governed by the balance of electron donors and terminal electron acceptors[5]. Lower $CH_4$ emissions in tidal vs. nontidal wetlands result from higher soil concentrations of sulfate, which acts as a terminal electron acceptor and allows sulfate-reducing bacteria to outcompete methanogenic communities for electron donors[5,6]. Site salinity, a proxy for sulfate availability, is the best-established predictor of $CH_4$ emissions from tidal wetlands, but it weakly constrains emission rates[6,7]. Overall, $CH_4$ emissions from tidal wetlands are extremely variable, and many sites emit $CH_4$ at rates that exceed C sequestration in terms of $CO_2$ equivalents[2,8,9]. Drivers of variability in $CH_4$ emissions other than sulfate are poorly understood[7,10]. Only few case studies have elucidated other important drivers of $CH_4$ emissions, such as sedimentation dynamics[11], organic matter quality and quantity[7], tidal pumping[12], and functional trait composition of plant communities[13–15]. Therefore, the consequences of perturbations on radiative forcing from tidal wetlands are difficult to predict and often unknown, currently representing one of the biggest challenges in blue C science[16].

Global change alters C sequestration and greenhouse-gas dynamics across ecosystems. In tidal wetlands, accelerated relative sea level rise (SLR) represents the overriding global change factor affecting ecosystem function in the long-term[17–19]. Although SLR poses a major threat to the stability of tidal wetlands, it also enhances their C stocks globally by stimulating C sequestration in soils[18,20]. SLR effects on tidal wetlands can therefore induce an important negative feedback to global warming[20]. Conversely, as SLR increases flooding frequency, leading to increasingly anaerobic soil conditions, it also yields the potential to stimulate $CH_4$ emissions. It is therefore possible that SLR-stimulated soil C sequestration is offset or even reversed by SLR stimulation of $CH_4$ emissions.

Methane emissions from nontidal wetland ecosystems often increase in response to global change factors such as elevated atmospheric levels of $CO_2$, rising temperatures, and eutrophication[21–25]. Stimulated $CH_4$ emissions in response to global change are often driven by the strong control of plant processes on soil $CH_4$ dynamics. Plants can stimulate $CH_4$ emissions from soils by increasing the input of organic matter serving as electron donors. Particularly, the input of recent photo-assimilates to the soil via root exudation is known to fuel methanogenic communities[5,26]. However, it is unclear if $CH_4$ responses to commonly studied global change factors in nontidal wetlands are transferable to tidal wetlands where SLR strongly interacts and often dominates other global change factors, modulating their effects on plant traits and microbial processes such as primary production and decomposition[18,27,28]. We therefore argue that the overriding control of SLR on tidal wetland functioning needs to be considered when estimating the effects of other global change drivers on $CH_4$ emissions. The effects of SLR on $CH_4$ emissions and the degree to which SLR modulates the effects of other global changes on $CH_4$ emissions has never been studied and cannot easily be projected. For instance, SLR-induced increases in flooding frequency are likely to exert opposing effects on the availability of two terminal electron acceptors that suppress methanogenesis, namely sulfate and oxygen. In addition, the relationship between sea level and electron donor availability (i.e. plant productivity) is not linear[29,30], further complicating projections of $CH_4$ dynamics in tidal wetlands.

Here we investigate the effects of SLR and its interactions with elevated atmospheric $CO_2$ and coastal eutrophication (i.e. elevated nitrogen levels) on $CH_4$ emissions from an estuarine tidal wetland. Multifactorial manipulations were implemented by applying a unique experimental design that combines field-deployed marsh mesocosms for sea level manipulation[31] and floating open top chambers to control atmospheric $CO_2$ concentrations[27]. Relationships observed in mesocosm studies were then tested against field data. We hypothesized that $CH_4$ emissions would increase in response to all factors—SLR, elevated $CO_2$, and eutrophication—and that SLR would be the dominant factor because of the strong control it exerts on oxygen availability. We predicted that $CH_4$ emissions would rise monotonically with SLR, and be greater within a given sea level when $CO_2$ or nitrogen were added as resources. We observed increases in $CH_4$ emissions in response to SLR and elevated $CO_2$, but not to eutrophication. SLR indeed exerted the strongest control on $CH_4$ emissions; however, its effect was nonlinear rather than monotonic, initially decreasing with SLR before increasing with SLR. This unexpected pattern in $CH_4$ emissions was primarily mediated by SLR-driven shifts in plant community composition that determined both the direction and magnitude of the $CH_4$ response. Subsequent in-situ observations confirmed that the same pattern occurs at the field-plot scale. Our findings therefore demonstrate that predictions of current and future greenhouse-gas dynamics of blue C ecosystems will require understanding of plant community dynamics and traits relevant to $CH_4$ cycling.

## Results and discussion

**Multiple global change effects on $CH_4$ emissions.** Global change treatments (sea level × nitrogen fertilization × elevated $CO_2$) were applied in a full-factorial design, and effects were analyzed using three-way (split plot) ANOVA[27] (Experiment 1). Sea level manipulations exerted the strongest effect on $CH_4$ emissions ($F = 10.78$; $p \leq 0.001$; Table 1; Fig. 1). The effect of relative sea level was nonlinear, counter to the expectation that increasing flooding will monotonically increase $CH_4$ emissions. $CH_4$ emissions were greatest at +40 cm above mean sea level (MSL; least-flooded elevation), show a steep drop from +40 cm to +20 cm above MSL, then increase from +20 cm to −5 cm (most-flooded elevation). Emissions from the least- and most-flooded elevations were not significantly different (Fig. 2a). Nonlinear regression analysis suggests a unimodal relationship between sea level and $CH_4$ emissions (log $CH_4$ emissions $_{(MSL)} = 0.001x^2 - 0.04x + 1.78$; $R^2 = 0.30$; $p \leq 0.001$; Supplementary Fig. 1).

The nitrogen fertilization treatment and any interactions thereof did not affect $CH_4$ emissions (all $F$ values ≤ 0.65; all $p$ values ≥ 0.59; Table 1; Fig. 1a). By contrast, an apparent $CO_2$ effect was indicated ($F = 5.84$; $p = 0.07$; Table 1), but likely masked to a certain degree by the overriding effect of the sea level treatment on our results. Indeed, two-way analyses within sea level treatments confirmed significant and strong stimulation of $CH_4$ emissions by elevated $CO_2$, with mean stimulation ranging from 70% at +20 cm to 670% at −5 cm relative to MSL (Fig. 1b).

**Table 1 Results of three-way split-plot ANOVA testing for effects of sea level, $CO_2$, and nitrogen treatments on $CH_4$ emissions ($n = 3$).**

| Factor | $F$ value | $p$ value |
|---|---|---|
| Sea level | **10.78** | **0.000** |
| Nitrogen | 0.05 | 0.833 |
| $CO_2$ | **5.84** | **0.073** |
| Sea level × Nitrogen | 0.65 | 0.590 |
| Sea level × $CO_2$ | 1.31 | 0.294 |
| Nitrogen × $CO_2$ | 0.12 | 0.728 |
| Sea level × Nitrogen × $CO_2$ | 0.40 | 0.753 |

Values are bold typed at $p \leq 0.10$.

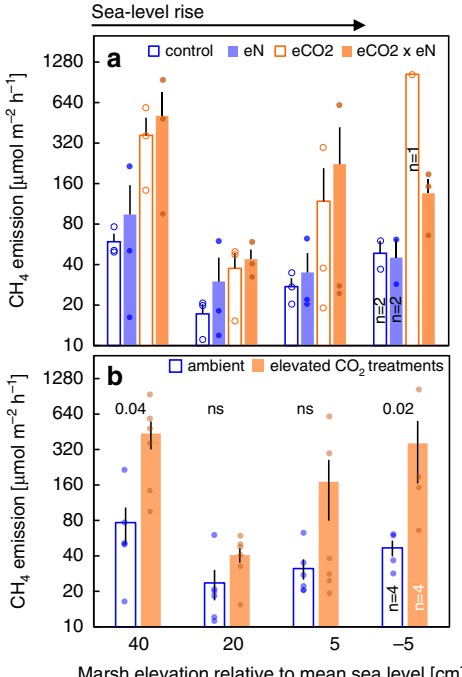

**Fig. 1 Interacting global change effects on CH$_4$ emissions. a** CH$_4$ emissions from field-based mesocosms of Experiment 1 exposed to four different sea level treatments (elevations relative to mean sea level) and full-factorial manipulations of nitrogen and CO$_2$ availability. (control = ambient nitrogen and ambient CO$_2$; eN = elevated nitrogen and ambient CO$_2$; eCO$_2$ = elevated CO$_2$ and ambient nitrogen; eCO$_2$ × eN = elevated CO$_2$ and elevated nitrogen). Data are presented as means ± SEM ($n = 3$ experimental units per group, based on mean values of duplicate mesocosms) and all datapoints are overlaid. Divergences from replication result from plant die-off at low elevations and are specified. **b** Ambient vs. elevated CO$_2$ treatments pooled for clearer illustration of eCO$_2$ effects. $P$ values above bars denote significant main effects ($p ≤ 0.05$) of the CO$_2$ treatment on CH$_4$ emissions (two-way ANOVA for each elevation separately). Data are presented as means ± SEM ($n = 6$ per group), divergences from replication are specified. The fraction of time flooded varied across the four sea level treatments as follows: 3–22–53–74%. Elevated CO$_2$ treatments were exposed to ambient [CO$_2$] + 300 ppm. Elevated N treatments received 25 g N m$^{-2}$ on a biweekly basis.

## Species shifts control global change effects on CH$_4$ emissions.

Experiment 1 was designed to examine the effects of interacting global change factors on plant growth in the context of inter-specific competition[27], and therefore global change treatments were applied to realistic plant assemblages, not single species. Plant responses of the two dominant species, the C4 grass *Spartina patens* (hereafter *Spartina*) and the C3 sedge *Schoenoplectus americanus* (hereafter *Schoenoplectus*), to sea level treatments reflected their abundance and biomass allocation along the natural elevation gradient and the SLR-driven encroachment of flooding tolerant *Schoenoplectus* into *Spartina* communities of the adjacent reference marsh and elsewhere[27,29,32–34] (Fig. 2b, compare Langley et al.[27] for a detailed presentation of plant biomass responses).

Here we found an unforeseen sharp decrease in CH$_4$ emissions with rising sea level in the higher parts of the tidal frame (Fig. 2a). This result was unexpected, because soil oxygen availability should have decreased as flooding duration increased from high to low elevations[27,35], simultaneously enhancing methanogenesis and suppressing methanotrophy. In the following we argue that the

observed decrease in CH$_4$ emissions was driven by a shift in species dominance from *Spartina*, dominant at high elevations of the marsh, to *Schoenoplectus*, dominant at low elevations (Fig. 2b).

CH$_4$ emissions were inversely related to *Schoenoplectus* aboveground biomass across all treatment combinations (log CH$_4$ emissions = $-0.0004x + 2.307$; $R^2 = 0.144$; $p ≤ 0.01$). Relationships between biomass parameters and CH$_4$ emissions were much stronger when restricted to certain CO$_2$- and nitrogen-treatment combinations. Specifically, CH$_4$ emissions showed the strongest negative relation to *Schoenoplectus* aboveground biomass within ambient CO$_2$-treatment combinations (Fig. 3), although similar but weaker relationships were also found under elevated CO$_2$ (Supplementary Fig. 2). The opposite response was observed in relation to *Spartina* aboveground biomass, which scaled positively with CH$_4$ emissions under ambient CO$_2$ (Fig. 3). Relationships between biomass parameters and CH$_4$ emissions were strongest when the dataset was restricted to the highest (least flooded) two treatments (+40 cm and +20 cm above MSL; Fig. 3), where changes in CH$_4$ emissions were most pronounced (Fig. 2a) and dominance of the two species was most balanced (Fig. 2b). Relationships of CH$_4$ emissions with plant parameters other than aboveground biomass were not significant, neither across nor within treatment groups (Supplementary Tables 1 and 2).

Plot-scale CH$_4$ data from the adjacent Smithsonian Global Change Research Wetland (GCReW) support the mesocosm results. Mean growing season CH$_4$ emissions were strongly related to the relative abundance of the two species (Fig. 4) and over three times greater from the higher elevation *Spartina*-dominated community of the marsh ($65 ± 37$ µmol m$^{-2}$ h$^{-1}$) than from the lower elevation *Schoenoplectus*-dominated community ($20 ± 5$ µmol m$^{-2}$ h$^{-1}$; $p ≤ 0.05$; $n = 3$). Both absolute CH$_4$ emission rates and differences induced by community composition correspond well to the findings of Experiment 1 (Fig. 1a, control treatment).

In order to evaluate the importance of these plant species-specific effects in mediating the relationship between sea level and CH$_4$ emissions in the upper tidal frame, a follow-up marsh organ experiment was conducted (Experiment 2). Experiment 2 did not use mixed species assemblages as in Experiment 1, but instead used pure communities of *Schoenoplectus* or *Spartina* to isolate species-level effects at two different sea levels. CH$_4$ emissions between the two species were dramatically different. Mean CH$_4$ emissions were 55 and 65 times greater from *Spartina* compared to *Schoenoplectus* at +15 cm and +35 cm above MSL, respectively ($F = 40.80$; $p ≤ 0.001$; Fig. 2c). Sea level ($F = 1.43$; $p = 0.26$) and the interaction of sea level and plant species ($F = 0.20$; $p = 0.66$) did not affect CH$_4$ emissions (Fig. 2c) demonstrating that CH$_4$ emissions as a function of sea level are primarily mediated by shifts in plant species composition, and that the direct (i.e. non-plant mediated) control of sea level on electron acceptor availability, such as oxygen, iron, and sulfate, is of less importance.

In contrast to the clear effects of sea level on *Spartina* vs. *Schoenoplectus* dominance in Experiment 1, CO$_2$ and nitrogen treatments did not induce significant shifts in species dominance within the mixed communities[27], demonstrating the stronger control of sea level than other global change factors on species composition. Both CO$_2$ and nitrogen treatments produced positive effects on plant biomass[27], but these did not translate into changes in CH$_4$ emissions. Nitrogen fertilization strongly and consistently increased *Schoenoplectus* and *Spartina* biomass across elevations[27] but had no effect on CH$_4$ emissions (Table 1 and Fig. 1a). Elevated CO$_2$ significantly increased *Schoenoplectus* and total aboveground biomass[27], two factors that were negatively related to CH$_4$ emissions (Fig. 3 and Supplementary Table 2), implying that the strong and positive effect of elevated CO$_2$ on CH$_4$ emissions (Fig. 1) was driven by plant processes that are not directly linked to

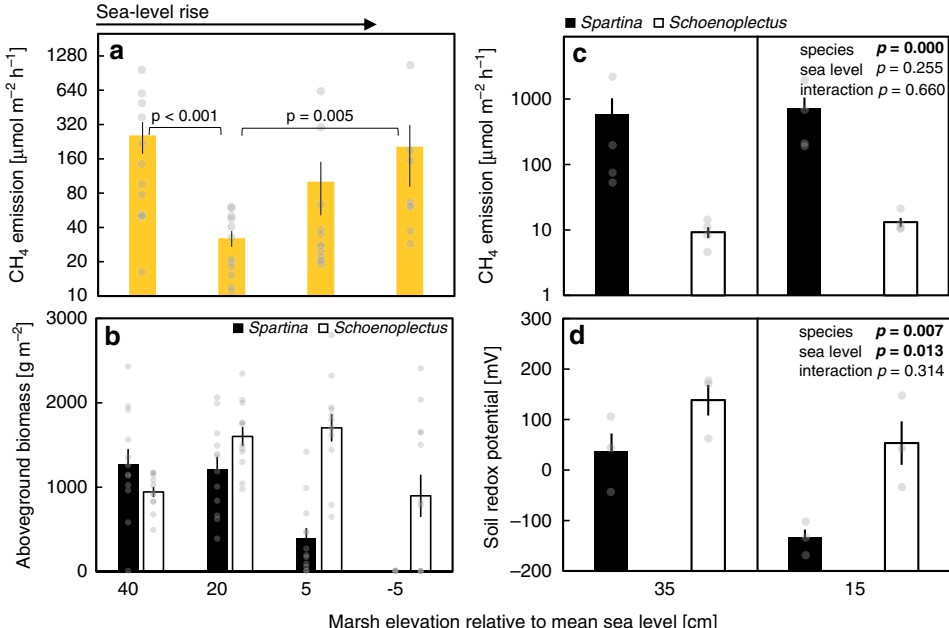

**Fig. 2 Plant species control on CH$_4$ emissions and soil redox. a** CH$_4$ emissions and **b** aboveground biomass of Experiment 1 mesocosms exposed to four different sea level treatments (elevations relative to mean sea level); all CO$_2$- and nitrogen-treatment combinations are pooled ($n = 12$ experimental units per group, based on means of duplicate mesocosms). Field-based mesocosms were planted with mixed communities of *Schoenoplectus americanus* and *Spartina patens*. **c** CH$_4$ emissions and **d** soil redox conditions (10 cm soil depth) of Experiment 2 mesocosms. Field-based mesocosms were either planted with *Schoenoplectus americanus* or *Spartina patens* and exposed to two different sea level treatments. CH$_4$ emissions were measured on $n = 4$ and redox on $n = 3$ mesocosms per group. All panels show means ± SEM plus an overlay of single datapoints. *P* values in **a** denote significant differences ($p \leq 0.05$) based on Tukey's HSD test. Two-way ANOVA results are shown in **c**, **d**. Data in **b** are redrawn after Langley et al.[27]. Data from *Schoenoplectus*-planted mesocosms in **c**, **d** are taken from Mueller et al.[50].

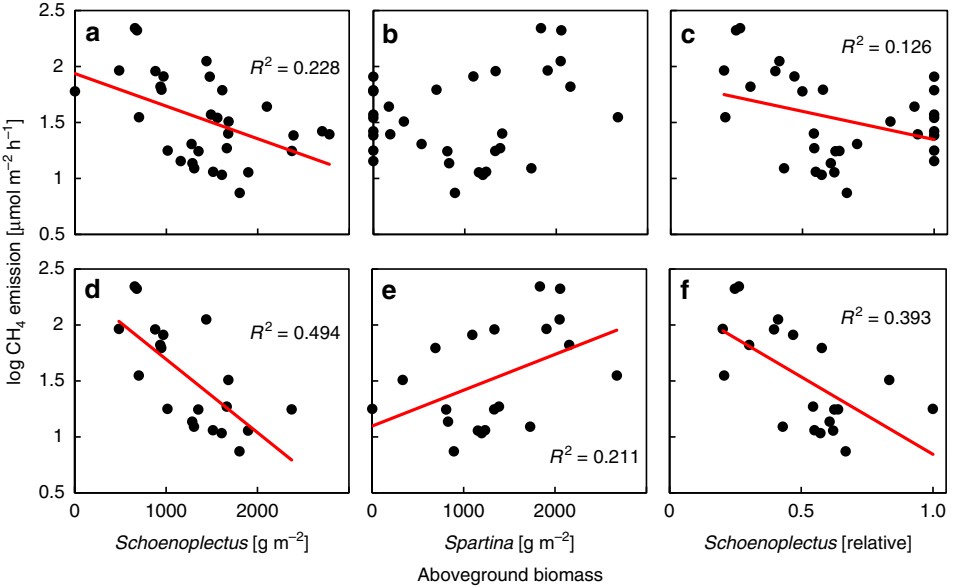

**Fig. 3 CH$_4$ emissions as function of aboveground biomass parameters.** Relationships between aboveground biomass and log CH$_4$ emissions within ambient CO$_2$ treatments of Experiment 1. **a**–**c** Relationships across all four sea level treatments ($n = 32$ mesocosms) and **d**–**f** across the two highest (least flooded) treatments (+40 cm and +20 cm above mean sea level; $n = 20$ mesocosms). Linear regression is shown for significant relationships ($p \leq 0.05$). Biomass data are taken from Langley et al.[27].

biomass. One likely process is the well-documented phenomenon of increased root exudation in response to elevated CO$_2$[36–39], acting as primary energy source for methanogenic communities[5]. In accordance with our findings, data from a long-term elevated CO$_2$ experiment in the adjacent GCReW field site show a strong CO$_2$ stimulation of CH$_4$ emissions from pure stands of *Schoenoplectus*[40]. Furthermore, elevated CO$_2$ increased both pore-water concentrations of CH$_4$ and dissolved organic C[41], effects that could likewise be attributed to greater inputs of organic matter via root exudation or rapid root turnover.

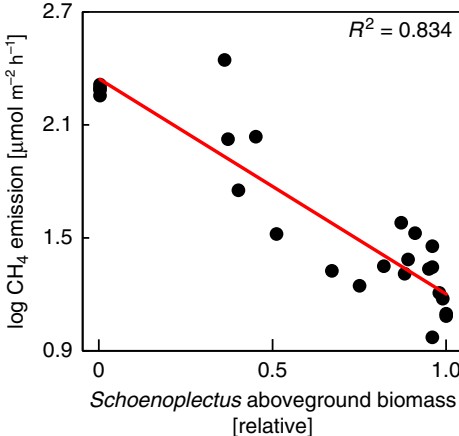

**Fig. 4 CH₄ emissions from field plots.** In-situ $CH_4$ emissions as a function of the aboveground biomass contributed by the C3 sedge *Schoenoplectus americanus* relative to total aboveground biomass, the remainder of which was C4 grasses (*Spartina patens* with small contributions of *Distichlis spicata*). $CH_4$ data are means of monthly in-situ flux measurements conducted during the 2019 growing season (Jun–Sep) across all ambient $CO_2$ plots of the *Salt Marsh Accretion Response to Temperature eXperiment* (SMARTX) ($n = 24$ field plots).

Previous work conducted at larger plot scales and over multiple years in mixed communities of the GCReW site has shown that elevated $CO_2$ and nitrogen fertilization shift the balance between *Schoenoplectus* and *Spartina* in opposite directions (i.e. nitrogen favored *Spartina* over *Schoenoplectus* and vice versa)[42]. Given the overriding control of plant community composition on $CH_4$ emissions found in the present study, this implies that the longer-term effects of these global change factors may differ from the effects presented here, which reflect relatively short-term effects over two growing seasons. However, the present work also demonstrates that SLR represents an overriding global change driver in the studied system. We therefore argue that shifts in plant species dominance in response to elevated $CO_2$ and nitrogen fertilization observed under ambient rates of SLR[42,43] may be less important under higher rates of SLR as simulated in the present study. This notion is supported by the observation that decadal-scale oscillations in local sea level at GCReW have stronger effects on plant community composition than elevated $CO_2$ and nitrogen fertilization treatments of the long-term field experiments[34,44].

**Plant traits affecting CH₄ dynamics**. In accordance with clear plant species effects on $CH_4$ emissions, soil redox conditions in the pure communities of Experiment 2 were more strongly affected by plant species than by sea level (Fig. 2d). Redox was markedly higher in *Schoenoplectus* vs. *Spartina* rhizospheres by c. 180 and 100 mV at +15 and +35 cm above MSL, respectively ($F = 13.0$; $p \leq 0.01$). Soil redox conditions reflect the balance between plant-mediated transport of electron donors and acceptors. Therefore, our findings demonstrate either a greater provision of electron acceptors (i.e. oxygen) or a lower provision of electron donors (organic matter) in *Schoenoplectus* vs. *Spartina* rhizospheres. Importantly, both mechanisms would cause lower $CH_4$ production in *Schoenoplectus* rhizospheres. Redox was significantly higher at +35 cm above MSL than at the lower and more frequently flooded +15 cm treatment ($F = 10.2$; $p \leq 0.05$), demonstrating the expected suppression of rising sea level on oxygen availability. Notably, there was no statistical difference ($p = 0.99$) in soil redox potential in the presence of *Schoenoplectus*

at the wettest treatment (+15 cm) and *Spartina* at the driest (+35 cm) treatment (Fig. 2d). Consistent with our $CH_4$ results, this demonstrates a stronger plant vs. sea level control on soil redox conditions in the studied system and underpins the primary control of plant species composition, and to a lesser degree sea level per se, on soil biogeochemistry.

The redox data suggest that greater $CH_4$ emissions in *Spartina* vs. *Schoenoplectus* are driven by plant traits affecting the balance between plant-mediated transport of electron donors and acceptors into the soil. There is abundant evidence to support greater supply of oxygen to the rhizosphere by *Schoenoplectus* vs. *Spartina* via root oxygen loss. Studies conducted on morphologically similar species of the same genus in tidal freshwater and nontidal wetland systems demonstrated markedly higher plant-stimulation of oxidation than production of $CH_4$[13,45–47]. Root oxygen loss by wetland plants supports higher rates of $CH_4$ oxidation and stimulates the decomposition of soil organic matter, a phenomenon called priming[48]. Previous work at the study site demonstrated high rates of priming in *Schoenoplectus* rhizospheres, whereas priming in *Spartina* rhizospheres was absent or even negative[49]. This finding provides further evidence of higher oxygen transport to soils by *Schoenoplectus* than *Spartina*, and it suggests opposing effects of root oxygen loss on priming and $CH_4$ emissions in a greenhouse-gas context. Indeed, in a past study we also demonstrated that priming in *Schoenoplectus* rhizospheres scales positively with aboveground biomass[50], opposite the response of $CH_4$ emissions to aboveground biomass in the present study (Fig. 3a, d).

The contrasting effects of the two species on $CH_4$ emissions may also be caused by differences in electron donor input, such as higher rates of root exudation in *Spartina* vs. *Schoenoplectus* rhizospheres. Recent studies in Chinese tidal wetlands demonstrated that invasive *Spartina alterniflora* stimulated $CH_4$ emissions through higher exudation of labile organic substrates from *S. alterniflora* roots in comparison to native species[15,51]. We do not have data on root exudate quality and quantity in *Spartina*- vs. *Schoenoplectus*-dominated mesocosms, but data from the adjacent reference marsh platform indeed show markedly higher porewater concentrations of dissolved organic C in *Spartina*[41,52].

One alternative explanation for greater $CH_4$ emissions from *Spartina* vs. *Schoenoplectus* is that *Spartina* supports greater rates of plant transport of $CH_4$ from the soil via the plant-aerenchyma system. This explanation, however, is implausible because *Spartina patens* has a poorly developed aerenchyma system compared to *Schoenoplectus americanus*[53], and concentrations of porewater $CH_4$ in the adjacent reference marsh are higher in *Spartina* vs. *Schoenoplectus* rhizospheres[52]. Taken together, it is likely that two processes—higher root oxygen loss by *Schoenoplectus* and higher root exudation by *Spartina*—explain the contrasting effects of these species on $CH_4$ emissions in the present study and thereby determined the dramatic change in $CH_4$ emissions in response to sea level-induced species shifts.

**Implications**. Other than salinity, drivers of variability in $CH_4$ emissions from tidal wetlands are poorly understood, which represents one of the biggest challenges to building robust numerical forecast models of greenhouse-gas dynamics for blue C ecosystems[16]. $CH_4$ emissions from the ambient $CO_2$ treatments of our main experiment ranged between 2.3 and 8.4 g $CH_4$ m⁻² year⁻¹ (Fig. 1b) and thereby reflect the lower spectrum of reported values for mesohaline marshes based on a recent global meta-analysis (−0.5 to 551.1 g $CH_4$ m⁻² year⁻¹)[7] and earlier work with focus on North America (3.3–32.0 g $CH_4$ m⁻² year⁻¹)[6].

Relative sea level exerted a strong, nonlinear control on $CH_4$ emissions. The difference between lowest and highest mean $CH_4$ emissions was 31 g $CH_4$ m$^{-2}$ year$^{-1}$ (Fig. 2a), corresponding to c. 6% of the total range of $CH_4$ emissions reported for tidal marshes globally[7] and to c. 95% of the total range reported for differences between meso- and polyhaline tidal marshes based on the salinity-$CH_4$ model of Poffenbarger et al.[6]. We furthermore show strong positive effects of elevated $CO_2$ which increased $CH_4$ emissions an amount similar to sea level effects. Our study thereby identifies two important drivers of $CH_4$ emissions both with a large potential to change the future greenhouse-gas balance of blue C ecosystems.

The main value of the present work is based on the mechanisms it illustrates, which are largely independent of absolute effect sizes. This is the first study to experimentally test if SLR interacts with other global change factors to change $CH_4$ emissions from blue C ecosystems. We demonstrate that predictions of both direction and magnitude of sea level effects on $CH_4$ emissions require an understanding of plant species traits that have the capacity to drive dramatic changes in redox chemistry. Furthermore, we show that effects of the global change factors elevated $CO_2$ and nitrogen interact differently with sea level. Effects of nitrogen fertilization were consistently null while the effects of elevated $CO_2$ were consistently positive. Indeed, $CO_2$ effects tended to amplify with more extreme sea levels. Our findings therefore yield important implications for modeling current and future greenhouse-gas dynamics of blue C ecosystems.

## Material and methods

**Study site**. The study was carried out in a tidal wetland site on Rhode river, a sub-estuary of the Chesapeake Bay in Maryland, USA (38°53′N, 76°33′W). The field site is home to the GCReW site operated by the Smithsonian Environmental Research Center. Tidal amplitude at the site is <50 cm and salinity generally <15 ppt. Soils are peats with organic matter contents >80%. Site vegetation is dominated by the C3 sedge *Schoenoplectus americanus* (hereafter *Schoenoplectus*) at lower, more frequently flooded elevations and by the C4 grass *Spartina patens* (hereafter *Spartina*) at higher, less frequently flooded elevations. The two species occur in pure and mixed communities depending on surface elevation. Over the past two decades, a fast, SLR-driven encroachment of *Schoenoplectus* into *Spartina* communities has been observed[34]. Plant growth at the site is nitrogen limited. Ammonium makes up >99% of the porewater inorganic nitrogen pool, and nitrate concentrations are usually below detection limits[42,54]. The main tidal creek of the GCReW site accommodates a marsh organ facility. Marsh organs (*sensu* Morris[31]) consist of field-based mesocosms arranged at different elevations, and thus different relative sea levels, to manipulate flooding frequency and assess the effects of accelerated relative SLR on plant and soil processes. Here we report on the results of two separate marsh organ experiments conducted between 2011 and 2012.

**Experimental designs**. The design of Experiment 1 has been described by Langley et al.[27] and was originally designed to study the effects of interacting global change factors on plant growth. It represents the first study to combine marsh organs and open top chambers to manipulate relative sea level and atmospheric $CO_2$ concentrations at the same time. An additional component of the study is an elevated nitrogen treatment. The three treatments were applied in a full-factorial design. Mesocosms (70-cm deep, 10-cm diameter) were filled with peat soil, planted with mixed native species assemblages of *Spartina* and *Schoenoplectus*, and evenly distributed on six separate marsh organs ($n = 24$ per

marsh organ). Initial planting reflected natural stem densities of the two species in the adjacent high mash[27]. Within each marsh organ, mesocosms were installed at the following six elevations in relation to MSL of the growing season (May–Sep): MSL −25 cm, MSL −15 cm, MSL −5 cm, MSL +5 cm, MSL +20 cm, and MSL +40 cm. Treatments covered the current relative sea level range of the adjacent marsh (three highest elevations) as well as future sea level scenarios (three lowest elevations)[27,54]. Long-term average SLR (90-year trend) at the site is c. 4 mm year$^{-1}$. MSL was calculated based on tide gauge data (Annapolis, MD, Station ID: 8575512, URL: https://tidesandcurrents.noaa.gov) after each growing season and could therefore only be estimated before mesocosm deployment. The fraction of time flooded ranged from 3% to 96% across the six elevations[27].

The elevated $CO_2$ treatment was applied by placing a floating open top chamber over each of the six marsh organs that was capable of rising and falling with the tide cycle. Three of the marsh organs were exposed to elevated $CO_2$ (ambient [$CO_2$] + 300 ppm, simulating an atmospheric $CO_2$ scenario projected for the year 2100[55]) by receiving additional $CO_2$ mixed into the air stream of a blower system connected to each open top chamber. The other three marsh organs were equipped with identical open top chambers and air blower systems but did not receive additional $CO_2$ via the air stream. Half of the mesocosms were exposed to an elevated nitrogen treatment projected to increase soil mineral nitrogen concentrations by c. 40%. Ammonium chloride solution equivalent to an nitrogen input of 25 g N m$^{-2}$ was injected to the rhizosphere on a biweekly basis throughout the growing season.

A follow-up marsh organ experiment, Experiment 2, was conducted to separate effects of plant species identity (i.e. *Schoenoplectus* vs. *Spartina*) from effects of interspecific plant competition on $CH_4$ emissions. This experiment used monocultures of either *Schoenoplectus* or *Spartina*, and no $CO_2$ or nitrogen treatments were applied. Mesocosms were exposed to three sea level treatments: MSL ±0 cm, MSL +15 cm, and MSL +35 cm. For details we refer the reader to Mueller et al.[50].

Mesocosm artifacts need to be considered when interpreting the absolute rates of $CH_4$ emissions and effect sizes reported here. For instance, marsh organ experiments at GCReW, including the present experiments, generally produce more biomass per area than the adjacent field site[27,34,43,49]. We therefore assessed the extent to which absolute $CH_4$ emissions and $CH_4$ emissions as a function of species composition (i.e. the key finding of our mesocosm experiments) differ between mesocosms and field plots of the adjacent marsh. Mean growing season $CH_4$ emissions were quantified in the *Salt Marsh Accretion Response to Temperature eXperiment* (SMARTX) operating in a high elevation, *Spartina*-dominated area and a low elevation, *Schoenoplectus*-dominated area of the adjacent marsh. A detailed description of the SMARTX study design is given by Noyce et al.[56]. Here we do not analyze temperature effects on $CH_4$ emissions, but compare $CH_4$ emissions from the ambient plots of the two plant communities ($n = 3$) and assess the relationship between the relative abundance of the two plant species and $CH_4$ emissions across all treatments ($n = 24$).

**Measurements**. $CH_4$ emission measurements followed the flux measurement protocol for marsh organs presented in Mueller et al.[50] with slight modifications for $CH_4$. In July 2011, in the second consecutive growing season of Experiment 1, mesocosms were carefully moved from the marsh organs into 120-L containers positioned directly adjacent. Due to poor plant survival at the lowest elevations, $CH_4$ emission measurements were restricted to elevations of MSL −5 cm and higher. Containers were filled

with creek water to the depth that corresponded to the water level that mesocosms were last exposed to in the marsh organ. Clear, acrylic flux chambers (volume = 7.5 L) were placed onto each mesocosm and sealed. Gas samples (20 mL) were collected from the chamber headspace every 20 min for a period of 2 h and analyzed for $CH_4$ using a gas chromatograph (Varian 450, Agilent Technologies). $CH_4$ fluxes were calculated from linear regression slopes (chamber headspace [$CH_4$] vs. time) following the ideal gas law, using chamber temperature for each given time point and assuming ambient pressure. Only fluxes with $R^2 \geq 0.8$ were used (mean $R^2 = 0.95 \pm 0.05$ SD, $N = 82$). The detection limit was 9 µmol $CH_4$ m$^{-2}$ h$^{-1}$.

$CH_4$ emission measurements of Experiment 2 were conducted in Sep 2012, after c. 4 months of plant growth in the marsh organ in the first growing season of the experiment. Sampling procedures followed Experiment 1, with the exception that samples were analyzed using a Shimadzu GC-14A (Shimadzu Corporation). Only fluxes with $R^2 \geq 0.8$ were used (mean $R^2 = 0.96 \pm 0.05$ SD, $N = 16$). The detection limit was 2 µmol $CH_4$ m$^{-2}$ h$^{-1}$. *Spartina* did not survive at MSL ±0 cm in Experiment 2. This elevation was therefore not considered for comparisons between species.

Field $CH_4$ emission measurements in SMARTX were conducted monthly from Jun 12 to Sep 4, 2019, 3 years after flux chamber bases were installed. Chambers ($40 \times 40 \times 40$ cm) were stacked onto each chamber base (total volume = 64–256 L) and covered with an opaque shroud. An ultra-portable greenhouse-gas analyzer (Los Gatos Research) was used to measure headspace $CH_4$ concentrations every 3 s for 5 min. Fluxes were calculated as described above and only fluxes that were significant at $p \leq 0.05$ were included in the analysis. Detection limit was <0.6 µmol $CH_4$ m$^{-2}$ h$^{-1}$.

In order to gain more mechanistic insight into potential effects of plant species shifts on $CH_4$ dynamics, soil redox conditions were measured in Experiment 2. Redox measurements were conducted during a single campaign in Sep 2012, after c. 4 months of plant growth in the marsh organ. Measurements were taken on $n = 3$ mesocosms per plant species and elevation at low tide. Three platinum-tipped redox electrodes[57] were inserted to a soil depth of 10 cm and allowed to equilibrate for 45 min. For readings, a calomel reference electrode (Fisher Scientific accumet) was inserted to a soil depth of 1 cm, and reference and redox electrodes were connected to a portable conductivity meter (Fisher Scientific accumet). Readings were corrected to the redox potential of the standard hydrogen electrode (+244 mV).

**Statistical analyses.** Analyses for Experiment 1 followed Langley et al.[27]. Three-way split-plot ANOVA was used to test for the effects of elevation (relative sea level), $CO_2$, nitrogen, and their factorial interactions on $CH_4$ emissions. Marsh organ (1–6) was included as a random factor in the model. Within single marsh organs, mesocosms of the same treatment combination were considered technical duplicates, and the mean of each duplicate was considered the experimental unit. Replication was therefore $n = 3$ per treatment. Subsequent two-way ANOVAs were used to assess $CO_2$ and nitrogen effects within each elevation treatment. Linear and nonlinear regression analysis was used to further explore the relationship of elevation and $CH_4$ emissions. In order to identify possible relationships between plant biomass parameters and $CH_4$ emissions, we used biomass data obtained from a destructive harvest in Sep 2011 (c. two months after the $CH_4$ emission measurements) that has been presented in Langley et al.[27]. Specifically, we conducted linear regression to test whether biomass parameters (Supplementary Table 2) and $CH_4$ emissions are related both across and within various treatment combinations. Two-way ANOVA was used to test for effects of plant species and elevation on $CH_4$ emissions and soil redox in Experiment 2. Tukey's HSD tests were used for pairwise comparisons following ANOVAs where appropriate. One-way ANOVA and linear regression were used to analyze the field $CH_4$ emission data (Fig. 4). $CH_4$ emission data typically show a log-normal distribution[40,58]. Data were log-transformed to improve normality (if required based on visual assessments) or when Levene's test indicated heterogenous variance. Regression analyses were conducted with both log-transformed and untransformed data. Analyses were conducted using R version 3.5.2 (R Foundation for Statistical Computing) and PAST version 3.20.[59]

**Reporting summary**. Further information on research design is available in the Nature Research Reporting Summary linked to this article.

## Data accessibility

Data used in this work are available from the corresponding authors upon request and at the Smithsonian Institution figshare repository (https://smithsonian.figshare.com) under the https://doi.org/10.25573/serc.12855323.

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

## Acknowledgements

This study was funded by Maryland Sea Grant (SA7528082, SA7528114-WW), the National Science Foundation Long-Term Research in Environmental Biology Program (DEB-0950080, DEB-1457100, and DEB-1557009), the US Department of Energy, Office of Science, Office of Biological and Environmental Research Program (DE-SC0014413 and DE-SC0019110), the Research Experience for Undergraduates (REU) Program (851303), and the Smithsonian Institution. Peter Mueller was supported by the DAAD (German Academic Exchange Service) PRIME fellowship program funded through the German Federal Ministry of Education and Research (BMBF). This study is a contribution to the Cluster of Excellence 'CLICCS—Climate, Climatic Change, and Society' and to the Center for Earth System Research and Sustainability (CEN) of Universität Hamburg.

## Author contributions

P.M. analyzed the data and wrote the initial manuscript. P.M. and L.R.A. conducted the marsh organ studies. G.L.N. conducted the field study and analyzed the data. J.P.M., T.J.M., and J.A.L. conceived the marsh organ studies. J.P.M. and G.L.N. conceived the field study. All authors contributed to manuscript editing.

## Funding

## Competing interests

The authors declare no competing interests.
