## [Peer Review File · Nature Communications]

REVIEWER COMMENTS

Reviewer #1 (Remarks to the Author):

The manuscript, "Plants determine methane response to sea level rise" presents an interesting global change experiment in an estuarine tidal wetland occupied by the C3 sedge *Schoenoplectus americanus* and the C4 grass *Spartina patens*. This work adds to a body of literature on blue carbon ecosystems, specifically the biogeochemical dynamics of estuarine tidal wetlands. This is the first study to experimentally test if sea level rise interacts with other global change factors to change CH₄ emissions from blue C ecosystems.

Although this work is not necessarily novel, it is of interest to a wide audience. The idea of plant species mediate methane emissions is not new (Matthews & Fung 1987; Whiting & Chanton 1992; Harriss et al., 1993; Cao et al., 1996; Joabsson et al., 1999). This study evaluated the degree to which tidal species influence CH₄ emission. These results are particularly interesting because the higher soil concentrations of sulfate introduced by the tidal influence should act as a terminal electron acceptor, allowing sulfate-reducing bacteria to outcompete methanogenic communities for electron donors. The tidal influence could potentially make methane emission negligible.

The results of this work suggest that CH₄ emissions are not negligible and there is a high sensitivity of CH₄ emissions to sea level rise and elevated atmospheric CO₂ concentrations in this estuarine tidal wetland. The results also show that CH₄ emissions are strongly influenced by the species present. Both the direction and magnitude of sea level effects on CH₄ emissions requires an understanding of plant species traits that have the capacity to drive dramatic changes in redox chemistry. This work has important implications for modeling the current and future greenhouse-gases of blue C ecosystems and their feedbacks to global warming.

While the manuscript is well written, it can be improved by adding a bit more detail on the global change factors and their influence on coastal wetland biogeochemical processes. Specifically, the introduction does not explain why or how global change factors may influence CH₄ emissions. Presenting hypotheses and what support for the hypothesis would look like would greatly improve this work. It is also not clear why the level of the global change factors used were chosen. It is also necessary to add detail to the analysis methods. What are the assumptions for the statistical approach used and how did you determine your data met these assumptions? What statistical package was used to analyze the data? Like many other studies, this work is limited by a small sample size and low temporal sampling intensity.

Additional suggestions:

Lines 51 – 54 is contrary to lines 32-41. It is not clear at this point why sulfate-reducing bacteria would not outcompete methanogenic communities.

Line 73: Consider changing the order of your treatments to reflect the order you discuss them: SLR x eN x eCO₂.

Line 92: This seems out of place because it is the first mention of plant growth. There has been no mention of the connection between CH₄ and primary productivity up until this point.

Lines 244. Consider adding the ambient CO₂ concentration in ppm and the elevated CO₂ concentration in ppm to the figure caption.

References

Cao, M., Marshall, S., & Gregson, K. (1996). Global carbon exchange and methane emissions from natural wetlands: Application of a process-based model. *Journal of Geophysical Research, D: Atmospheres*, 101(D9), 14399–14414.

Harriss, R., Bartlett, K., Frohking, S., & Crill, P. (1993). Methane Emissions from Northern High-Latitude Wetlands. In R. S. Oremland (Ed.), *Biogeochemistry of Global Change: Radiatively Active Trace Gases Selected Papers from the Tenth International Symposium on Environmental Biogeochemistry*, San Francisco, August 19–24, 1991 (pp. 449–486). Springer US.

Joabsson, A., Christensen, T. R., & Wallén, B. (1999). Vascular plant controls on methane emissions from northern peatforming wetlands. *Trends in Ecology & Evolution*, 14(10), 385–388.

Matthews, E., & Fung, I. (1987). Methane emission from natural wetlands: Global distribution, area, and environmental characteristics of sources. *Global Biogeochemical Cycles*, 1(1), 61–86.

Whiting, G. J., & Chanton, J. P. (1992). Plant-dependent CH₄ emission in a subarctic Canadian fen. *Global Biogeochemical Cycles*, 6(3), 225–231.

Reviewer #2 (Remarks to the Author):

The authors present findings from a multifactorial mesocosm experiment in an estuarine tidal marsh in order to investigate global change effects on CH₄ fluxes with sea level rise under four scenarios/treatments: ambient air (control), elevated CO₂, elevated nitrogen levels, and elevated CO₂ + elevated N. They find that shifts in CH₄ emissions with sea level rise was driven primarily by plant community composition. They also find that elevated CO₂ stimulated CH₄ fluxes, independent from sea level rise.

The results of the two experiments have been published elsewhere. Additional data (CH₄ fluxes) are presented here and combined with previous findings. The manuscript is well written and the data support the conclusion.

The following comments are indented to help to improve the manuscript.

Title: The title could be more specific to better reflect the two main findings. For example, the authors could consider to include the terms “plant community composition”, “tidal wetlands”, “tidal marsh” and “elevated CO₂”. Understandably, the authors tried to find a short and catchy title but after reading the manuscript, I find that the title doesn’t reflect the main two findings well enough.

Abstract: I think the abstract could be expanded. It would be good to include a couple of sentences on the methods (e.g. mention the four treatments). If word limit is an issue, the introduction could be slightly cut.

Introduction: paragraph 1 (L27-38): Please define “tidal wetlands”. The term is blurry. Blue carbon systems or as the authors state “tidal wetlands and seagrass meadows” include mangrove forest. Either include mangroves here, or mention only marsh and remove seagrass. When introducing methane from non-tidal wetlands the publications by Poulter et al. 2017 and Saunio et al. 2019 should be included (most recent methane emissions from inland wetlands). Blue carbon burial offsets by methane emissions have been published by Rosentreter et al. 2018, Al-Haj and Fulweiler 2020 and Oreska et al. 2020. Salinity is an important control on methane production in coastal sediments. However, the statement that methane emissions are negligible for salinities >18 should be re-considered. Several studies have shown that methane fluxes in coastal wetlands can be significant at high salinities (see Al-Haj and Fulweiler 2020, Rosentreter et al. 2018) driven e.g. by groundwater inputs, tidal pumping (e.g. Maher et al. 2013, Call et al. 2014).

L36: Can the authors list a few other potential drivers – even if not well understood?

Methods: Can you specify what is meant by “marsh organ”?

L353-356: Were gas samples collected at the exact same time from each chamber or was there a time

delay? The authors sampled the headspace of the chamber every 20 min over 2 hours (n=6). How much gas was sampled (volume?) and was the air that was sampled replaced in the chamber? What is the r² of the regression slopes used for flux calculations? Were all regressions linear? (If there were any spikes there might have been some ebullition). What is the accuracy/detection limit of the gas chromatograph? What is the sampling error associated with the CH₄ concentration and fluxes? More details needed here.

It would be helpful to have access to the original data.

Section: Multiple global change effects on CH₄ emissions (L66-83)

As the authors mention in the introduction, methane is highly variable in coastal systems. For a better understanding of the dataset, it would be recommended to use boxplots that show the data distribution with mean, median and range instead of bar plots. This may be particularly important as the number of methane fluxes is moderate (exp1 n=44; exp 2 n = 16).

Figure S1 clearly shows the CH₄ fluxes in the four treatments and gives a good overview of the dataset. It may be preferred to show Figure S1 in the main manuscript instead of Figure 1.

Figure 1 the black bar: is this the combined data of eCO₂ and eCO₂ + eN (n=6) versus control (n=3)? If so, why is at -5cm the control n=4? This does not match up with what is presented in Fig S1 where at -5cm the control is n=2. Please, clarify.

Were there any epiphytic algae or benthic algae present in the mesocosms? This could be another important factor to consider here when discussing shifts in plant community composition.

The authors state that the nitrogen fertilization treatment had no effect on methane emissions, yet Fig S1 shows eN is higher than the control at 40, 20 and 5 cm. Even if this is statistically not significant, this might be a trend worth mentioning? Again, it would be good to see boxplots.

Section: Species shifts control global change effects on CH₄ emissions (L85-155)

The sharp decrease of methane from 40 to 20cm is indeed surprising. The authors argue that the shift in species dominance was responsible for this decrease, which is supported by Fig 2B and 2C (although unfortunately no *Spartina* at -5cm). I wonder what the below-ground biomass of the two marsh species is? Are there any estimates or available data? Methane is produced primarily in anaerobic sediments and during the process of carbon burial. Below-ground biomass could be an important parameter for marsh burial (Chmura et al. 2003), hence methane production in sediments.

I am missing the comparison of methane fluxes in this study with other marsh studies (of the same or different climate zone). Are the reported fluxes at the higher or upper end for tidal marsh? How do the methane fluxes compare to other micro-tidal and meso-haline tidal marshes? Some more context would be helpful.

Refs:

Poulter, B., P. Bousquet, J. G. Canadell, and others. 2017. Global wetland contribution to 2000–2012 atmospheric methane growth rate dynamics. *Environ. Res. Lett.* 12: 094013. doi:10.1088/1748-9326/aa8391

Saunio, M., A. R. Stavert, B. Poulter, and others. 2019. The global methane budget 2000-2017. *Earth Syst. Sci. Data* preprint.

Chmura, G. L., S. C. Anisfeld, D. R. Cahoon, and J. C. Lynch. 2003. Global carbon sequestration in tidal, saline wetland soils. *Global Biogeochem. Cycles* 17: 1111. doi:10.1029/2002GB001917

Al-Haj, A. N., and R. W. Fulweiler. 2020. A synthesis of methane emissions from shallow vegetated

coastal ecosystems. *Glob. Chang. Biol.* 2017: gcb.15046. doi:10.1111/gcb.15046

Rosentreter, J. A., D. T. Maher, D. V Erler, R. H. Murray, and B. D. Eyre. 2018. Methane emissions partially offset "blue carbon" burial in mangroves. *Sci. Adv.* 4: eaao4985. doi:10.1126/sciadv.aao4985

Oreska, M. P. J., K. J. McGlathery, L. R. Aoki, A. C. Berger, P. Berg, and L. Mullins. 2020. The greenhouse gas offset potential from seagrass restoration. *Sci. Rep.* 10: 7325. doi:10.1038/s41598-020-64094-1

Maher, D. T., I. R. Santos, L. Golsby-Smith, J. Gleeson, and B. D. Eyre. 2013. Groundwater-derived dissolved inorganic and organic carbon exports from a mangrove tidal creek: The missing mangrove carbon sink? *Limnol. Oceanogr.* 58: 475–488. doi:10.4319/lo.2013.58.2.0475

Call, M., D. T. Maher, I. R. Santos, and others. 2015. Spatial and temporal variability of carbon dioxide and methane fluxes over semi-diurnal and spring-neap-spring timescales in a mangrove creek. *Geochim. Cosmochim. Acta* 150: 211–225. doi:10.1016/j.gca.2014.11.023

Reviewer #3 (Remarks to the Author):

Mueller et al. present results from two marsh organ experiments evaluating changes in methane fluxes in response to (1) the singular and interactive effects of elevated CO₂, sea level, and nitrogen loading; and (2) plant species identity and sea level. Results from the first experiment demonstrated nonlinear changes in CH₄ fluxes in response to increasing sea levels and CO₂ concentration. Interestingly they observed that CH₄ fluxes were correlated with the shifting dominance of the grass *Spartina patens*, at higher elevations, and the sedge *Schoenoplectus*, at lower elevations. Correlations between CH₄ fluxes and sedge aboveground biomass were significant, but explained a small fraction of the variance (L101). Experiment 2 followed on these results and convincingly demonstrated higher CH₄ fluxes and lower redox potential in *Spartina*-only vs. sedge-only treatments, regardless of sea level treatment. Combined the results from the two experiments highlight the complexity of interactions between physical drivers (sea level), ecological drivers (plant identity), and anthropogenic disturbances (CO₂ levels) in affecting marsh carbon cycling (as CH₄ fluxes). The experiments and results are timely and important steps in predicting how marsh carbon budgets and radiative forcing may change under different conditions.

The authors conducted multi-factorial experiments but it is not always clear if data in the figures reflect select treatments or combinations of treatments (e.g., Fig 1 and S1). I found S1 to be much clearer than Figure 1 and would suggest using S1 in the main text rather than the current Figure 1.

Comparisons between CH₄ fluxes and certain subsets of treatments (Fig. 3, S3, L100-111) were also confusing. These seem to highlight higher correlation values when the dataset was restricted to fewer and fewer treatments. Figures 2, 3, and S3 seem to only contain a restricted amount of plant data. However, it may be that all of the data are presented but some of the treatments are combined. This ambiguity creates confusion and makes it difficult to evaluate the results and follow the text.

Additional rationale for the ecological basis of the treatments is needed, particularly since some results differed between the present study and long-term, larger scale field studies (L144-155). How do the SL treatments relate to current and predicted levels of SLR? I.e., what is the projected time period and / or emissions scenario when the +40 cm treatment would occur? Do environments currently at +40 cm or +20 cm have *Spartina* and *Schoenoplectus*? Do these two species often co-occur or is one competitively dominant? Biomass of the plants was reported per mesocosm (10 cm diameter) and not per m². It would be helpful to normalize biomass to area for comparison to the nearby field site (and

other sites). At what density and biomass were the grass and sedge initially planted in both experiments? How do initial and final biomasses compare, within the experiments and to the field?

Were data collected continuously throughout the experiment or from the end only (e.g., gas fluxes, redox, plant height, etc)?

Along the same lines, some discussion about potential artifacts of marsh organ experiments and how those might affect translation to the field environment would be useful. There are some fairly strong statements about the generality of the results (L20-22) and more should be done to establish that the manipulated sea level treatments, fluxes, biomass, etc. are consistent with field conditions or observations. There are some comparisons to the field – mainly the GCRew sites – but additional support would strengthen the arguments presented.

It would be helpful to include the frequency of flooding or % time of inundation along side the SL axis. Is the +40 cm treatment flooded once a season or more frequently? Is the -5cm treatment continuously submerged?

The authors measured plant biomass and CO₂ fluxes and I was wondering if biomass – specific productivity rates might be useful in understanding patterns in CH₄ fluxes. This could be interesting particularly under the different CO₂ and N treatments, and if there were shifts in above or below ground plant biomass or C:N ratios. I.e., do higher CO₂ levels lead to greater productivity, higher C:N ratios, and root DOC exudates. Were there relationships between CO₂-based GEP or NEP or R rates with CH₄ fluxes?

Since the grass is a C₄ plant and the sedge is C₃, I was also wondering if the authors could explore their priming discussion further, by presenting δ¹³C data from the CH₄ (and CO₂?) fluxes. The experiments were conducted a while ago so this may not be a possibility.

Additional, more specific comments are included in the attached PDF.

Dear Editor,

We appreciate the thoughtful comments provided by the three reviewers and reply to each separately in the following. The most significant improvement to our manuscript represents the inclusion of original CH₄-emission data of the 2019 growing season from our field site. This was done to address important reviewer comments concerning the applicability of our marsh-organ results. Dr. Genevieve Noyce, responsible for the CH₄-emission measurements at the Smithsonian Global Change Research Wetland, contributed these data and joined the author team.

(author responses in red)

REVIEWER COMMENTS

Reviewer #1 (Remarks to the Author):

The manuscript, "Plants determine methane response to sea level rise" presents an interesting global change experiment in an estuarine tidal wetland occupied by the C3 sedge *Schoenoplectus americanus* and the C4 grass *Spartina patens*. This work adds to a body of literature on blue carbon ecosystems, specifically the biogeochemical dynamics of estuarine tidal wetlands. This is the first study to experimentally test if sea level rise interacts with other global change factors to change CH₄ emissions from blue C ecosystems. Although this work is not necessarily novel, it is of interest to a wide audience. The idea of plant species mediate methane emissions is not new (Matthews & Fung 1987; Whiting & Chanton 1992; Harriss et al., 1993; Cao et al., 1996; Joabsson et al., 1999). This study evaluated the degree to which tidal species influence CH₄ emission. These results are particularly interesting because the higher soil concentrations of sulfate introduced by the tidal influence should act as a terminal electron acceptor, allowing sulfate-reducing bacteria to outcompete methanogenic communities for electron donors. The tidal influence could potentially make methane emission negligible. The results of this work suggest that CH₄ emissions are not negligible and there is a high sensitivity of CH₄ emissions to sea level rise and elevated atmospheric CO₂ concentrations in this estuarine tidal wetland. The results also show that CH₄ emissions are strongly influenced by the species present. Both the direction and magnitude of sea level effects on CH₄ emissions requires an understanding of plant species traits that have the capacity to drive dramatic changes in redox chemistry. This work has important implications for modeling the current and future greenhouse-gases of blue C ecosystems and their feedbacks to global warming.

While the manuscript is well written, it can be improved by adding a bit more detail on the global change factors and their influence on coastal wetland biogeochemical processes. Specifically, the introduction does not explain why or how global change factors may influence CH₄ emissions. Presenting hypotheses and what support for the hypothesis would look like would greatly improve this work.

Hypotheses have been developed (59-62, 68-73, 79-86).

It is also not clear why the level of the global change factors used were chosen. It is also necessary to add detail to the analysis methods.

Rationale for the global change treatments has been specified (391-92, 311-13, 421-422), and more detail has been added to the analysis methods (453-65).

What are the assumptions for the statistical approach used and how did you determine your data met these assumptions? What statistical package was used to analyze the data? Like many other studies, this work is limited by a small sample size and low temporal sampling intensity.

We followed the same statistical approach as in our previous study on the biomass response (same experiment; Langley et al. 2013). More detail concerning assumptions of the statistical procedure has been added (502-506). We included information on the statistical software used. We agree with the reviewer's remark concerning the low sample size and temporal sampling intensity (resulting from logistical constraints), which affects our statistical power and the applicability of absolute values (effect sizes). However, we believe that the main value of our work is in the mechanisms it reveals, which are largely independent of absolute effect sizes (259-60). Furthermore, we included mean growing season CH₄-emission data from the adjacent field site to provide additional support for the strong plant-species control on CH₄ emissions, the central finding of our study (139-45, Figure 4).

Additional suggestions:

Lines 51 – 54 is contrary to lines 32-41. It is not clear at this point why sulfate-reducing bacteria would not outcompete methanogenic communities.

We agree with the reviewer. We clarify by developing hypotheses and acknowledging that the effect of SLR on the biogeochemistry of tidal wetlands is complex and difficult to predict based on our current understanding (68-73).

Line 73: Consider changing the order of your treatments to reflect the order you discuss them: SLR x eN x eCO₂.

We changed the order of the global change treatments in Table 1 to reflect the order of the discussion/results presentation.

Line 92: This seems out of place because it is the first mention of plant growth. There has been no mention of the connection between CH₄ and primary productivity up until this point.

This has been stated more clearly now in the introduction (59-62).

Lines 244. Consider adding the ambient CO₂ concentration in ppm and the elevated CO₂ concentration in ppm to the figure caption.

We specified the treatments in Figure 1 caption.

Reviewer #2 (Remarks to the Author):

The authors present findings from a multifactorial mesocosm experiment in an estuarine tidal marsh in order to investigate global change effects on CH₄ fluxes with sea level rise under four scenarios/treatments: ambient air (control), elevated CO₂, elevated nitrogen levels, and elevated CO₂ + elevated N. They find that shifts in CH₄ emissions with sea level rise was driven primarily by plant community composition. They also find that elevated CO₂ stimulated CH₄ fluxes, independent from sea level rise. The results of the two experiments have been published elsewhere. Additional data (CH₄ fluxes) are presented here and combined with previous findings. The manuscript is well written and the data support the conclusion. The following comments are indented to help to improve the manuscript.

Title: The title could be more specific to better reflect the two main findings. For example, the authors could consider to include the terms “plant community composition”, “tidal wetlands”, “tidal marsh” and “elevated CO₂”. Understandably, the authors tried to find a short and catchy title but after reading the manuscript, I find that the title doesn’t reflect the main two findings well enough.

We agree with the reviewer and changed the title accordingly.

Abstract: I think the abstract could be expanded. It would be good to include a couple of sentences on the methods (e.g. mention the four treatments). If word limit is an issue, the introduction could be slightly cut.

A brief methods summary has been added to the abstract.

Introduction: paragraph 1 (L27-38): Please define “tidal wetlands”. The term is blurry. Blue carbon systems or as the authors state “tidal wetlands and seagrass meadows” include mangrove forest. Either include mangroves here, or mention only marsh and remove seagrass.

We rephrased the sentence accordingly (32).

When introducing methane from non-tidal wetlands the publications by Poulter et al. 2017 and Saunio et al. 2019 should be included (most recent methane emissions from inland wetlands). Blue carbon burial offsets by methane emissions have been published by Rosentreter et al. 2018, Al-Haj and Fulweiler 2020 and Oreska et al. 2020. Salinity is an important control on methane production in coastal sediments. However, the statement that methane emissions are negligible for salinities >18 should be re-considered. Several studies have shown that methane fluxes in coastal wetlands can be significant at high salinities (see Al-Haj and Fulweiler 2020, Rosentreter et al. 2018) driven e.g. by groundwater inputs, tidal pumping (e.g. Maher et al. 2013, Call et al. 2014).

All these are very useful suggestions. The references have been added (33, 40). The statement on negligible CH₄ emissions from sites >18 ppt has been revised (37-38)

L36: Can the authors list a few other potential drivers – even if not well understood?

We list more potentially important drivers (40-42).

Methods: Can you specify what is meant by “marsh organ”?

We specify the term marsh organ as follows (395-98):

“The main tidal creek of the GCRew site accommodates a marsh organ facility. Marsh organs (sensu Morris¹⁹) consist of field-based mesocosms arranged at different elevations, and thus different relative sea levels, to manipulate flooding frequency and assess the effects of accelerated relative SLR on plant and soil processes.”

L353-356: Were gas samples collected at the exact same time from each chamber or was there a time delay?

Gas samples of Exp 1 were taken over six days (1 day per marsh organ) and sampling day (marsh organ) was included as a random factor in the ANOVA model (compare summary report document). Gas samples of Exp 2 were taken on a single day.

The authors sampled the headspace of the chamber every 20 min over 2 hours (n=6). How much gas was sampled (volume?) and was the air that was sampled replaced in the chamber? What is the r² of the regression slopes used for flux calculations? Were all regressions linear? (If there were any spikes there might have been some ebullition). What is the accuracy/detection limit of the gas chromatograph? What is the sampling error associated with the CH₄ concentration and fluxes? More details needed here.

Each sample was 20 mL (total sample volume = 120 mL). The sample volume was not replaced. However, the total headspace of the flux chambers was >7500 mL, so that sampling did not have a relevant effect on the headspace pressure.

Ebullition was only observed in mesocosms without vegetation (die-off due to excessive flooding). Regressions were linear. Details on regression slopes and other details are now given in the methods (458-465).

It would be helpful to have access to the original data.

All data will be made publicly available upon acceptance of the manuscript. We can also provide the data to the reviewer immediately upon request.

Section: Multiple global change effects on CH₄ emissions (L66-83)

As the authors mention in the introduction, methane is highly variable in coastal systems. For a better understanding of the dataset, it would be recommended to use boxplots that show the data distribution with mean, median and range instead of bar plots. This may be particularly important as the number of methane fluxes is moderate (exp1 n=44; exp 2 n = 16).

See next comment (combined response)

Figure S1 clearly shows the CH₄ fluxes in the four treatments and gives a good overview of the dataset. It may be preferred to show Figure S1 in the main manuscript instead of Figure 1.

We added Figure S1 to the main manuscript (new Figure 1a) and kept former Figure 1 (new Figure 1b) for a clearer presentation of the CO₂ effect size. Because the number of observations is as low as 2 and 1 in some of the groups in Figure 1a, we believe that the use of boxplots would reduce clarity. Instead we included the new Figure S1, depicting single observations, to provide a better picture of the data distribution.

Figure 1 the black bar: is this the combined data of eCO₂ and eCO₂ + eN (n=6) versus control (n=3)? If so, why is at -5cm the control n=4? This does not match up with what is presented in Fig S1 where at -5cm the control is n=2. Please, clarify.

The number resulted from 2 controls + 2 eN x aCO₂ resulting in 4 observations under ambient CO₂ concentrations. Combining former Figures S1 and 1 in the new version of the manuscript should help clarify.

Were there any epiphytic algae or benthic algae present in the mesocosms? This could be another important factor to consider here when discussing shifts in plant community composition.

We have no data on the abundance of microphytobenthos of the mesocosm soil surface. Epiphytic algae have not been observed.

The authors state that the nitrogen fertilization treatment had no effect on methane emissions, yet

Fig S1 shows eN is higher than the control at 40, 20 and 5 cm. Even if this is statistically not significant, this might be a trend worth mentioning? Again, it would be good to see boxplots.

This is a good point that did not occur to use before. Indeed, showing the total range and distribution of single observations (new Figure S1) makes clear why N effects are not significant: The higher mean values of the N-treatment groups result from few high values.

Section: Species shifts control global change effects on CH₄ emissions (L85-155)

The sharp decrease of methane from 40 to 20cm is indeed surprising. The authors argue that the shift in species dominance was responsible for this decrease, which is supported by Fig 2B and 2C (although unfortunately no *Spartina* at -5cm). I wonder what the below-ground biomass of the two marsh species is? Are there any estimates or available data? Methane is produced primarily in anaerobic sediments and during the process of carbon burial. Below-ground biomass could be an important parameter for marsh burial (Chmura et al. 2003), hence methane production in sediments.

We did not include sufficient information on the relationship of belowground biomass and CH₄ emissions, which has been addressed now. There were no significant relationships between any of the belowground biomass parameters assessed (AG:BG: ratio, *Schoenoplectus* rhizome biomass, *Spartina* rhizome biomass, root depth distribution, fine root biomass) and CH₄ emissions. This was not clear from the old version of our manuscript. We now explicitly state this (136-37) and include extra material in the supplement (Table S1, S2). If anything, there is a negative correlation between *Schoenoplectus* rhizome biomass and CH₄ emissions ($r = -0.33$, $p = 0.09$). A finding that supports the negative effect of *Schoenoplectus* aboveground biomass on CH₄ emissions.

I am missing the comparison of methane fluxes in this study with other marsh studies (of the same or different climate zone). Are the reported fluxes at the higher or upper end for tidal marsh? How do the methane fluxes compare to other micro-tidal and meso-haline tidal marshes? Some more context would be helpful.

Comparisons to other systems have been included (244-248; 251-52).

Refs:

Poulter, B., P. Bousquet, J. G. Canadell, and others. 2017. Global wetland contribution to 2000–2012 atmospheric methane growth rate dynamics. *Environ. Res. Lett.* 12: 094013. doi:10.1088/1748-9326/aa8391

Saunois, M., A. R. Stavert, B. Poulter, and others. 2019. The global methane budget 2000-2017. *Earth Syst. Sci. Data* preprint.

Chmura, G. L., S. C. Anisfeld, D. R. Cahoon, and J. C. Lynch. 2003. Global carbon sequestration in tidal, saline wetland soils. *Global Biogeochem. Cycles* 17: 1111. doi:10.1029/2002GB001917

Al-Haj, A. N., and R. W. Fulweiler. 2020. A synthesis of methane emissions from shallow vegetated coastal ecosystems. *Glob. Chang. Biol.* 2017: gcb.15046. doi:10.1111/gcb.15046

Rosentreter, J. A., D. T. Maher, D. V. Emler, R. H. Murray, and B. D. Eyre. 2018. Methane emissions partially offset “blue carbon” burial in mangroves. *Sci. Adv.* 4: eaao4985. doi:10.1126/sciadv.aao4985

Oreska, M. P. J., K. J. McGlathery, L. R. Aoki, A. C. Berger, P. Berg, and L. Mullins. 2020. The greenhouse gas offset potential from seagrass restoration. *Sci. Rep.* 10: 7325. doi:10.1038/s41598-020-64094-1

Maher, D. T., I. R. Santos, L. Golsby-Smith, J. Gleeson, and B. D. Eyre. 2013. Groundwater-derived dissolved inorganic and organic carbon exports from a mangrove tidal creek: The missing mangrove carbon sink? *Limnol. Oceanogr.* 58: 475–488. doi:10.4319/lo.2013.58.2.0475

Call, M., D. T. Maher, I. R. Santos, and others. 2015. Spatial and temporal variability of carbon dioxide and methane fluxes over semi-diurnal and spring-neap-spring timescales in a mangrove creek. *Geochim. Cosmochim. Acta* 150: 211–225. doi:10.1016/j.gca.2014.11.023

Reviewer #3 (Remarks to the Author):

Mueller et al. present results from two marsh organ experiments evaluating changes in methane

fluxes in response to (1) the singular and interactive effects of elevated CO₂, sea level, and nitrogen loading; and (2) plant species identity and sea level. Results from the first experiment demonstrated nonlinear changes in CH₄ fluxes in response to increasing sea levels and CO₂ concentration. Interestingly they observed that CH₄ fluxes were correlated with the shifting dominance of the grass *Spartina patens*, at higher elevations, and the sedge *Schoenoplectus*, at lower elevations. Correlations between CH₄ fluxes and sedge aboveground biomass were significant, but explained a small fraction of the variance (L101). Experiment 2 followed on these results and convincingly demonstrated higher CH₄ fluxes and lower redox potential in *Spartina*-only vs. sedge-only treatments, regardless of sea level treatment. Combined the results from the two experiments highlight the complexity of interactions between physical drivers (sea level), ecological drivers (plant identity), and anthropogenic disturbances (CO₂ levels) in affecting marsh carbon cycling (as CH₄ fluxes). The experiments and results are timely and important steps in predicting how marsh carbon budgets and radiative forcing may change under different conditions.

The authors conducted multi-factorial experiments, but it is not always clear if data in the figures reflect select treatments or combinations of treatments (e.g., Fig 1 and S1). I found S1 to be much clearer than Figure 1 and would suggest using S1 in the main text rather than the current Figure 1.

This is a very good point. We added Figure S1 to the main manuscript (new Figure 1a) but kept former Figure 1 (new Figure 1b) for a clearer presentation of the CO₂ effect.

Comparisons between CH₄ fluxes and certain subsets of treatments (Fig. 3, S3, L100-111) were also confusing. These seem to highlight higher correlation values when the dataset was restricted to fewer and fewer treatments.

We argue that a higher correlation when less sea-level treatments are included results from the increasingly lower degree of abiotic variability. Across the entire dataset, CH₄ emissions would be affected by the sea-level treatment independent of changes in plant species composition. We decided to present the subset of observations restricted to the two highest elevations (Figure 2 D-F) for two reasons: First, it is between these two elevations where the unexpected dramatic change in CH₄ emissions occurred, and we are here presenting evidence that this change is driven by a shift in species composition. Second, relevant abundances of both species were restricted to these two elevations, whereas the two lower elevations support very little or no growth of *Spartina*. As a consequence, inclusion of the two lower elevations introduces variability to the regression model that cannot be explained by species composition. We additionally decided to test for the same correlations identified in Figure 2 E-F within the highest elevation to prove that species composition controls CH₄ emissions independent of changes in flooding frequency. We agree with the reviewer that this reduces clarity, and we moved these panels (G-J) to the supplement (Figure S4).

Figures 2, 3, and S3 seem to only contain a restricted amount of plant data. However, it may be that all of the data are presented but some of the treatments are combined. This ambiguity creates confusion and makes it difficult to evaluate the results and follow the text.

General response: All biomass data we refer to has been published in an earlier paper focused on the biomass response as part of the same experiment (Langley et al. 2013). We are only re-drawing those biomass data here that are most useful for the interpretation of the CH₄-emission data. For the remainder we can only provide reference to the original publication. We acknowledge that this should have been clearer throughout the manuscript, and we now provide a clearer statement when the biomass data is first introduced (109-116):

“Experiment 1 was designed to examine the effects of interacting global change factors on plant growth

*in the context of interspecific competition²⁶, and therefore global change treatments were applied to realistic plant assemblages, not single species. Plant responses of the two dominant species, the C4 grass *Spartina patens* (hereafter *Spartina*) and the C3 sedge *Schoenoplectus americanus* (hereafter *Schoenoplectus*) to sea level treatments reflected their abundance and biomass allocation along the natural elevation gradient and the SLR-driven encroachment of flooding tolerant *Schoenoplectus* into *Spartina* communities of the adjacent reference marsh and elsewhere^{26,28,31–33} (Figure 2B, compare Langley et al.¹⁷ for a detailed presentation of plant biomass responses)."*

We furthermore refer the reader to Langley et al. in all figure captions using biomass data.

Specific response: The reviewer is correct. Figures 2 and S3 show all biomass data, but all treatments have been combined. This was done to understand the effects of plant biomass parameters and CH₄ emissions across treatments. We also tested for effects of biomass parameters within sea-level-, N-, and CO₂-treatment combinations, but only present the relevant results of these analyses. Specifically, biomass effects within sea-level treatments are presented in Table S1. We furthermore observed that biomass effects on CH₄ emissions were clearest when compared within the CO₂ treatments separately. Figure 3 depicts these results for the aCO₂ treatment, whereas Figure S3 shows the results of the eCO₂ treatment.

Additional rationale for the ecological basis of the treatments is needed, particularly since some results differed between the present study and long-term, larger scale field studies (L144-155). How do the SL treatments relate to current and predicted levels of SLR? I.e., what is the projected time period and / or emissions scenario when the +40 cm treatment would occur?

We added more detail on CO₂ and sea-level treatments to the respective section of the methods (391-92; 411-13; 421-22).

Sea-level treatments captured the current elevation range of the marsh vegetation (+40 – +5 cm) and an additional lower treatment positioned 10 cm lower (-5 cm). Long-term average SLR (90-yr trend) at the site is c. 4 mm/yr. When (and if) the -5-cm-treatment scenario is going to occur depends on plant-sea-level interactions that we do not fully understand, yet.

It is important to note, however, that SLR effects operate across the entire elevational gradient of the ecosystem. Our study design is not restricted to a single SLR treatment. For instance, to understand SLR effects on the highest elevated communities (+40 cm), we need to compare them with the (+20 cm) treatment. In terms of CH₄ emissions, we see a strong decrease from +40 to 20 cm. In lower parts of the elevational gradient the CH₄ response to SLR is reversed.

Do environments currently at +40 cm or +20 cm have *Spartina* and *Schoenoplectus*? Do these two species often co-occur or is one competitively dominant?

MSL was calculated after each growing season (for each season separately) and does not represent a fixed datum. This was done to enable comparisons between Exp 1 and 2. However, based on the absolute elevations of our mesocosms, patterns of species dominance are well reflected by our sea-level treatments (compare Langley et al. 2013 and Mozdzer et al. 2016 for detail).

The two species are structured along the elevational gradient with *Spartina* being dominant at higher, less frequently flooded elevations and *Schoenoplectus* at lower elevations. Mixed communities of the two species exist at intermediate elevations. Over the past 2 decades, SLR was driving the encroachment of *Schoenoplectus* into *Spartina* communities and increased its abundance in mixed communities of the GCREW site (Drake 2014) (compare Figures below).

[does not correspond to elevation above MSL as given in the present study]

Correlation between sea level and increase of *Schoenoplectus* biomass in mixed communities. Symbols indicate CO₂ treatments.

Figures from Drake (2014).

We added these points to different parts in the methods (391-92) and results/discussion (114-15).

Biomass of the plants was reported per mesocosm (10 cm diameter) and not per m². It would be helpful to normalize biomass to area for comparison to the nearby field site (and other sites). At what density and biomass were the grass and sedge initially planted in both experiments? How do initial and final biomasses compare, within the experiments and to the field?

We normalized biomass data (Figure 2 b, Figure 3). The two species were planted at densities reflecting natural stem densities of the two species in the adjacent high marsh (~500 stems m⁻² for *Schoenoplectus*; ~3000 stems m⁻² for *Spartina* (White et al., 2012; Shepard 2010) based on half of the mesocosm surface area (39.3 cm²) in Exp. 1 and total surface area of Exp. 2 (78.5 cm²) (compare Langley et al. 2013 for detail). This information has been added to the methods (407-8).

Greater standing aboveground biomass than in the marsh platform is one commonly observed artifact of the marsh organ experiments conducted at GCRW. For instance, mean aboveground biomass of Exp.1 was 2360 ± 665 g/m², which is clearly higher than published values from the adjacent marsh, commonly ranging between 300 and 1100 g/m² (e.g. Langley et al 2009; Langley and Megonigal 2010; Drake 2014). However, biomass can vary by factor 2-3 between years, and maximum values >2000 g/m² have been observed in high-biomass years (Adam Langley and Pat Megonigal, unpublished data, see figure below).

Unpublished data from the field CO₂/N experiment at GCRew.
AG biomass in 30 x 30 cm quadrats sampled between 2005 and 2019 (Langley and Megonigal, unpublished).

We acknowledge that potential marsh organ artifacts on biomass production need to be considered when interpreting absolute rates of CH₄ emissions.

*“Mesocosm artifacts need to be considered when interpreting absolute rates of CH₄ emissions and effect sizes reported here. For instance, marsh organ experiments at GCRew – including the present – generally produce more biomass per area than the adjacent field site^{26,33,42,48}. We therefore compared CH₄ emissions from the marsh organ with mean growing season CH₄ emissions from the Salt Marsh Accretion Response to Temperature eXperiment (SMARTX) operating in a high elevated, *Spartina*-dominated area and a low elevated, *Schoenoplectus*-dominated area of the the adjacent marsh. A detailed description of the SMARTX study design is given by Noyce et al.⁵⁵.”* 434-440

The main value of our work is in the mechanisms it illustrates, not absolute rates and effect sizes. Still, rates of CH₄ emissions of Exp 1 agree well with field data from GCRew (Figure 4). More importantly, the additional field data provide support for the strong plant-species control on CH₄ emissions, the central finding of our study (139-45, Figure 4).

Were data collected continuously throughout the experiment or from the end only (e.g., gas fluxes, redox, plant height, etc)?

Along the same lines, some discussion about potential artifacts of marsh organ experiments and how those might affect translation to the field environment would be useful. There are some fairly strong statements about the generality of the results (L20-22) and more should be done to establish that the manipulated sea level treatments, fluxes, biomass, etc. are consistent with field conditions or observations. There are some comparisons to the field – mainly the GCRew sites – but additional support would strengthen the arguments presented.

CH₄, redox, and plant parameters were assessed only once. Compare lines 449, 476.

We therefore compared CH₄ emissions from the marsh organ with mean growing season CH₄ emissions from the adjacent field site (Figure 4).

It would be helpful to include the frequency of flooding or % time of inundation alongside the SL axis. Is the +40 cm treatment flooded once a season or more frequently? Is the -5cm treatment continuously submerged?

We added that information to the methods (416-17) and the caption of Figure 1. The +40 cm treatment was flooded 3% of the time (more than 24 h / month). The -5 cm treatment was flooded 78% of the time.

The authors measured plant biomass and CO₂ fluxes and I was wondering if biomass – specific productivity rates might be useful in understanding patterns in CH₄ fluxes. This could be interesting particularly under the different CO₂ and N treatments, and if there were shifts in above or below ground plant biomass or C:N ratios. I.e., do higher CO₂ levels lead to greater productivity, higher C:N ratios, and root DOC exudates. Were there relationships between CO₂-based GEP or NEP or R rates with CH₄ fluxes?

Agree, this would certainly be interesting. Unfortunately, those data are not available.

Since the grass is a C₄ plant and the sedge is C₃, I was also wondering if the authors could explore their priming discussion further, by presenting δ¹³C data from the CH₄ (and CO₂?) fluxes. The experiments were conducted a while ago so this may not be a possibility.

¹³C-CH₄ data are not available. However, compare Mueller et al. (2016) for a discussion of SLR effects on priming based on ¹³C-CO₂.

Additional, more specific comments are included in the attached PDF (copied from pdf)

46 SLR would presumably increase inundation by sulfate-rich waters. A shift to methanogenesis would imply that sulfate became limiting. This is potentially more likely in low porosity peaty soils and less likely in higher porosity soils with greater percolation.

We agree. This would depend on several factors, i.e. [sulfate]/salinity of floodwater and substrate availability. We now clarify by developing our hypotheses more carefully based on R1's comments (68-73).

69 This is surprising because in the ambient treatments, it appears that there is a clear SL effect in the 40 cm treatment compared to the others. There also appears to be a SLR x CO₂ interaction, but that turned out non-significant. Did the statistics include all of the treatments or a subset (CO₂ and SL only?).

The SL effect is consistent in both eCO₂ and aCO₂. CH₄ emissions are highest from the lowest and highest elevation and significantly different compared to 20 cm (Figure 1 b, Figure 2 a).

Table 1 shows results of the 3-way ANOVA (all treatment combinations included). We also tested for CO₂ effects in subsets of the data (CO₂ and SL only), which also turned out non-significant (SL x CO₂: $p = 0.215$). Overall, we agree with the reviewer, however, our design provides insufficient statistical power to detect this potential CO₂ x SL interaction.

103 Relationship

Thanks, addressed.

128 These data do not seem to be in any of the figures. Please add them.

130 It is unclear why some plant biomass data are reported here but others are not. It seems that that plant and gas flux data are from the same experiment so showing both would be helpful.

Comment has been addressed above.

132 This is unclear. Do the authors mean variability in plant biomass or shifts in the abundances of each species across treatments? If the latter, then more explanation would be useful.

We referred to biomass. We have rewritten this paragraph for clarity (165-67)

Suppl Table. 134 This table reports relationships between biomass and log CH₄. It is not broken out by treatment (CO₂, N)

We added the required information to the table and rewrote the respective lines in the discussion to improve clarity (165-67).

135 What about productivity? Could CO₂ data provide insight into relationships between biomass-specific productivity rates (and possibly oxygen translocation, root exudation, etc) and CH₄ fluxes

It certainly would. Unfortunately, we don't have those data.

146 Why were the same results not found here? Which plant increased under which condition?

This was stated unclearly in the previous version. We specified (178-79).

149 It would be helpful to have this earlier. Does this reflect both experiments? If experiment 1 only then it is clear that plant species had an important effect on CH₄ fluxes, but not an overriding one:

figure 3 shows that species identity had relatively small explanatory power across the entire data set (0-23%). The results of experiment 2 are clearer.

We now clarified in the methods that flux measurements were conducted after 2 seasons in Exp.1 and 1 season in Exp.2. The reviewer certainly brings up an interesting point. However, we agree only in part. That is, the effect driven by species composition in Exp.1 is mainly determined by the change between the +40 cm and the +20 cm treatment. Here, plant aboveground biomass explains almost 50% of the variability in CH₄ emissions. Relevant abundances of both species were restricted to these two elevations, whereas the two lower elevations support very little to no growth of *Spartina*. Therefore, inclusion of the two lower elevations introduces variability to the regression model that cannot be explained by species composition (compare earlier comment concerning Figure 3). The new field data we added (Figure 4) underscores the overriding control of plants.

170 Repetitive

deleted

177 Were belowground biomass and rooting depth similar between these 2 plants?

We do not have comprehensive species-specific belowground biomass for our main experiment (Exp 1). (There is however species-specific rhizome biomass compare comment below). In our field site, *Spartina* has higher total belowground biomass and a higher root density in the topsoil (5-10 cm) than *Schoenoplectus*, whereas *Schoenoplectus* has higher root density at depth and roots significantly deeper than *Spartina* (Saunders et al. 2007). These data are also reflected in the data of Exp. 2. However, belowground biomass was harvested in the second season of this experiment, one year after the CH₄ investigations, and was therefore not considered in this paper.

In any case, it is important to note that belowground biomass parameters (depth distribution, fine root mass, rhizome mass) were unrelated to CH₄ emissions (for clarity we included the new Table S1). Therefore, we do not think that CH₄ emissions were strongly controlled by rooting depth distribution or biomass parameters. The discussion following line 211 of our manuscript is therefore focused on physiological rhizosphere traits (i.e. root oxygen loss, root exudation).

Fig 1 Are these same data as in S1, but with treatments combined?

Yes. Please note that, based on the comments provided by R1+2, Figure S1 is now part of Figure 1.

Fig 1 Please add information regarding flooding frequency or percent time inundated to the axis or text.

Comment has been addressed above.

306 The elevations at which each plant is found relative to MSL would be useful for comparison to the experimental treatments.

Comment has been addressed above.

What was the initial mix that was planted (e.g., 50:50 each species)? Was the mixture based on biomass or another variable?

Comment has been addressed above.

323 What set of predictions are the sea level treatments meant to mimic?

Comment has been addressed above.

382 Was species specific belowground biomass used in regressions?

Comment partly addressed above. For Exp. 1, we have species specific rhizomes biomass. Roots could not be visually distinguished. Rhizomes were distinguished for a subset of samples. Rhizome biomass represented the by far largest fraction of belowground biomass irrespective of the sea level treatment. CH₄ emissions tended to decrease with increasing *Schoenoplectus* rhizome mass, reflecting the *Schoenoplectus* aboveground-biomass effect on CH₄ emissions. We added this information to the supplement (Table S2).

REVIEWERS' COMMENTS:

Reviewer #1 (Remarks to the Author):

The manuscript, "Plants determine methane response to sea level rise" presents an interesting global change experiment in an estuarine tidal wetland occupied by the C3 sedge *Schoenoplectus americanus* and the C4 grass *Spartina patens*. This work adds to a body of literature on blue carbon ecosystems, specifically the biogeochemical dynamics of estuarine tidal wetlands. This is the first study to experimentally test if sea level rise interacts with other global change factors to change CH₄ emissions from blue C ecosystems. This work is of interest to a wide audience and shows the degree to which tidal species influence CH₄ emission. These results are particularly interesting because it shows that CH₄ emissions are not negligible and there is a high sensitivity of CH₄ emissions to sea level rise and elevated atmospheric CO₂ concentrations in this estuarine tidal wetland. The results also show that CH₄ emissions are strongly influenced by the species present. Both the direction and magnitude of sea level effects on CH₄ emissions requires an understanding of plant species traits that have the capacity to drive dramatic changes in redox chemistry. The authors have made significant improvements to this work. I offer a few minor changes:

Line 15: Add "(C)" after carbon.

Line 19: remove brackets around CO₂ and format subscript.

Line 22: remove brackets around CO₂ and format subscript.

Line 26: remove brackets around CO₂.

Line 28: remove brackets around CO₂.

Line 458: Capitalize r² to match figures.

Line 464: Capitalize r² to match figures.

Line 488: Remove #

Reviewer #2 (Remarks to the Author):

The points raised by the reviewers have been satisfactorily addressed. I only have a few minor comments.

L33 Add "Microbial" CH₄ production...

L40-41: groundwater and/or porewater inputs (tidal pumping) is a well-known control on surface water methane concentration and fluxes in tidal wetland. Might be worth mentioning here and later in the discussion.

L59 "Stimulated CH₄ emissions in response to global change" repeat of previous sentence

L463 delete "of"

Reviewer #3 (Remarks to the Author):

The authors revisions have substantially improved the manuscript. The edits have made the manuscript clearer and more informative.

The negative and positive relationships between *Schoenoplectus* and *Spartina* with CH₄ emissions, respectively, are clear. However the correlations explain a relatively small proportion of the variance, particularly across all SLR treatments ($r^2 < 0.5$, Fig. 3). Along the same lines, the relationships

between species identity and biomass and CH₄ emissions in experiment 1 are not clear cut. For instance, CH₄ emissions are similar at the +40 cm SL treatment, where *Schoenoplectus* and *Spartina* biomasses are similar, and at the -5 SL treatments, where only *Schoenoplectus* is present (Fig. 2). And, in experiment 2, "there was no statistical difference in soil redox potential in the presence of *Schoenoplectus* at the wettest treatment (+15 cm) and *Spartina* at the driest (+35 cm) treatment (p = 0.991) (Figure 2D)." (L202)

Taken together the results do demonstrate that plant species influence CH₄ emissions and soil conditions, and that these effects are as strong as, or stronger than, the experimental global change disturbances. However, a complementary interpretation could be that plant species-specific effects – on CH₄ emissions and soil redox - differ with disturbance severity (e.g., SL) and are modified by other factors such as species interactions. There are strongly worded statements throughout (e.g., L155-157, 204-205) with more moderated wording in the conclusions (L263). Given the complexity of the results, and the scale of the experiment, the manuscript would be strengthened by capturing some of this nuance throughout.

Further clarity about the statistics in experiment 2 would be useful. This experiment included single species treatments at 2 SL's. It is reported that there was not an interaction between species and SL (L154) and it is unclear how this was detected since the design was not fully factorial.

Abstract: consider adding in plant species treatment information in the methods description.

Dear Editor,

We appreciate the thoughtful comments provided by the three reviewers and reply to each separately in the following.

(author responses in green)

Reviewer #1 (Remarks to the Author):

The manuscript, "Plants determine methane response to sea level rise" presents an interesting global change experiment in an estuarine tidal wetland occupied by the C3 sedge *Schoenoplectus americanus* and the C4 grass *Spartina patens*. This work adds to a body of literature on blue carbon ecosystems, specifically the biogeochemical dynamics of estuarine tidal wetlands. This is the first study to experimentally test if sea level rise interacts with other global change factors to change CH₄ emissions from blue C ecosystems. This work is of interest to a wide audience and shows the degree to which tidal species influence CH₄ emission. These results are particularly interesting because it shows that CH₄ emissions are not negligible and there is a high sensitivity of CH₄ emissions to sea level rise and elevated atmospheric CO₂ concentrations in this estuarine tidal wetland. The results also show that CH₄ emissions are strongly influenced by the species present. Both the direction and magnitude of sea level effects on CH₄ emissions requires an understanding of plant species traits that have the capacity to drive dramatic changes in redox chemistry. The authors have made significant improvements to this work. I offer a few minor changes:

Line 15: Add "(C)" after carbon.

Line 19: remove brackets around CO₂ and format subscript.

Line 22: remove brackets around CO₂ and format subscript.

Line 26: remove brackets around CO₂.

Line 28: remove brackets around CO₂.

Line 458: Capitalize r² to match figures.

Line 464: Capitalize r² to match figures.

Line 488: Remove #

→All comments have been addressed.

Reviewer #2 (Remarks to the Author):

The points raised by the reviewers have been satisfactorily addressed. I only have a few minor comments.

L33 Add "Microbial" CH₄ production...

L40-41: groundwater and/or porewater inputs (tidal pumping) is a well-known control on surface water methane concentration and fluxes in tidal wetland. Might be worth mentioning here and later in the discussion. → Citation Call et al. (2015) has been added.

L59 "Stimulated CH₄ emissions in response to global change" repeat of previous sentence

L463 delete "of"

→All comments have been addressed.

Reviewer #3 (Remarks to the Author):

The authors revisions have substantially improved the manuscript. The edits have made the manuscript clearer and more informative.

The negative and positive relationships between *Schoenoplectus* and *Spartina* with CH₄ emissions, respectively, are clear. However, the correlations explain a relatively small proportion of the

variance, particularly across all SLR treatments ($r^2 < 0.5$, Fig. 3). Along the same lines, the relationships between species identity and biomass and CH₄ emissions in experiment 1 are not clear cut. For instance, CH₄ emissions are similar at the +40 cm SL treatment, where *Schoenoplectus* and *Spartina* biomasses are similar, and at the -5 SL treatments, where only *Schoenoplectus* is present (Fig. 2). And, in experiment 2, “there was no statistical difference in soil redox potential in the presence of *Schoenoplectus* at the wettest treatment (+15 cm) and *Spartina* at the driest (+35 cm) treatment ($p = 0.991$) (Figure 2D).” (L202). Taken together the results do demonstrate that plant species influence CH₄ emissions and soil conditions, and that these effects are as strong as, or stronger than, the experimental global change disturbances. However, a complementary interpretation could be that plant species-specific effects – on CH₄ emissions and soil redox - differ with disturbance severity (e.g., SL) and are modified by other factors such as species interactions. There are strongly worded statements throughout (e.g., L155-157, 204-205) with more moderated wording in the conclusions (L263). Given the complexity of the results, and the scale of the experiment, the manuscript would be strengthened by capturing some of this nuance throughout. Further clarity about the statistics in experiment 2 would be useful. This experiment included single species treatments at 2 SL’s. It is reported that there was not an interaction between species and SL (L154) and it is unclear how this was detected since the design was not fully factorial.

→The design of Exp. 2 was fully (two)-factorial. For clarity, we included two-way anova results in figure 2c-d and detailed in the figure caption. We believe that missing clarity about the two-factorial nature of Exp.2 made it hard to follow our argument that plant effects were stronger than sea-level effects (e.g. previous L155-57). We also addressed the wording in L204-205.

Abstract: consider adding in plant species treatment information in the methods description.

→Information has been added